## PERSPECTIVE

# More than myocytes: The heart's electrical ensemble

Jana Grune[1,2], Christof Lenz[3,4], Clément Cochain[5,6] and Noor Momin[7,8] (ORCID)

[1]*Department of Cardiothoracic and Vascular Surgery, Deutsches Herzzentrum der Charité (DHZC), Berlin, Germany*

[2]*Charité – Universitätsmedizin Berlin, Corporate member of Freie Universität Berlin and Humboldt-Universität zu Berlin, Institute of Physiology, Berlin, Germany*

[3]*Department of Clinical Chemistry, University Medical Center Göttingen, Göttingen, Germany*

[4]*Bioanalytical Mass Spectrometry Group, Max Planck Institute for Multidisciplinary Sciences, Göttingen, Germany*

[5]*Université Paris Cité, Inserm, PARCC, Paris, France*

[6]*Institute of Experimental Biomedicine, University Hospital Würzburg, Würzburg, Germany*

[7]*Department of Bioengineering, University of Pennsylvania, Philadelphia, PA, USA*

[8]*Center for Precision Engineering for Health, University of Pennsylvania, Philadelphia, PA, USA*

Email: jana.grune@charite.de

Handling Editors: Bjorn Knollmann & T Alexander Quinn

The peer review history is available in the Supporting Information section of this article (https://doi.org/10.1113/JP289679#support-information-section).

### The heart's multicellular ensemble

When the heart beats, what is really happening? For decades, the answer seemed straightforward: specialized cells in the heart called cardiomyocytes contract in synchronized waves, enabling the heart to function as a biological pump. This cardiomyocyte-centric view dominated cardiac biology, with researchers focusing almost exclusively on how these cardiomyocytes process and propagate electrical signals.

Two reviews in this issue of *The Journal of Physiology* challenge this simplified model: Simon-Chica et al. (2026) and Wu et al. (2026) reveal how the heart operates as an electrical network involving multiple cell types, including fibroblasts, immune cells and others. Together, these papers, alongside others, highlight a far more complex cellular ensemble underlying the most basic physiological function of the heart.

### Changing keys in cardiac electrophysiology

Both reviews describe how the heart contains other cells amidst cardiomyocytes that actively participate in cardiac electrical conduction. They describe two distinct types of electrical coupling between these cells: gap junctions and ephaptic coupling. Gap junctions act like tiny electrical cables directly connecting cells, allowing ion currents to flow between them. Ephaptic coupling describes how cells influence each other's electrical activity by creating local electrical fields in the narrow extracellular spaces between them.

Both computational models and experimental findings converge on the finding that these modes of electrical coupling are used by cardiac fibroblasts and immune cells such as macrophages to actively support cardiomyocytes' electrical activity. These cells can speed up, slow down or even block electrical conduction, fundamentally altering the heart's ability to function in both health and disease.

Beyond fibroblasts and immune cells, other non-cardiomyocytes in the heart also contribute to its electrical activity. Endothelial cells can also couple to cardiomyocytes via gap junctions, but whether such coupling occurs meaningfully in native tissue remains unclear. Additionally, the heart possesses many neurons, especially in arrhythmic niches, yet how neurons couple to cardiomyocytes is not well understood. Understanding the role of all cardiac cell types on cardiac electrophysiology in more detail is important.

### Modern instruments to decode multicellular diversity and electrophysiology in the heart

Our current understanding of multicellular contribution to conduction emerged through a convergence of cutting-edge technologies. Electron microscopy has revealed cell-to-cell connections at nanometre scales, exposing the intricate electrical highways between different cardiac cell types that were previously hidden from view. Genetic engineering approaches such as optogenetics allowed researchers to control specific cell types with light, turning fibroblasts or immune cells on and off to test their electrical contributions in living hearts. Additionally, sophisticated computational models have helped determine the contribution of different ions flowing through gap junctions and ephaptic coupling and even predict how cells interact electrically.

Although these approaches have revolutionized the field, they also highlight the technical challenges that remain. Genetic targeting even with modern promoters to drive the expression of optogenetic tools can lack desired cell-type specificity needed to investigate different non-cardiomyocytes cell subsets. Computational models rely on assumed parameters that are sometimes difficult to empirically measure. Refinement in transgenic tools and implementation of artificial intelligence and machine learning into cardiac electrophysiology modelling promises to overcome some of these technical barriers and reveal even deeper insights into the heart's electrical complexity.

Looking forward, multi-omics approaches, such as comprehensive analyses of genes (transcriptomics) and proteins (proteomics), can be used to catalogue the stunning cellular diversity within the heart. Single cell transcriptomics (Rizzo et al., 2023) and sophisticated proteomics approaches (Brandenburg et al., 2022) have and will continue to offer insights into the distinct electrical properties of cell types and proteins.

### Future orchestration of the electrical assemble using bioengineering

Understanding heterocellular coupling opens entirely new therapeutic avenues (Fig. 1). Current anti-arrhythmic drugs primarily target cardiomyocytes at the same time as ignoring the broader cellular network that participates in conduction.

Instead of just modulating cardiomyocytes, future treatments could also target the fibroblasts and immune cells within scarred or inflamed tissue exhibiting poor conduction. Advanced bioengineering approaches are making this vision reality.

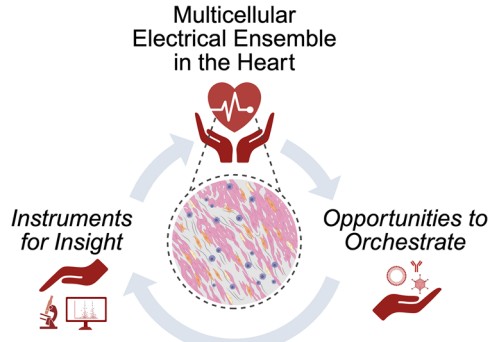

**Figure 1. Overview**

The multicellular electrical ensemble present in the heart is being deciphered by modern instruments and will inform new therapeutic avenues to treat arrhythmia and conduction disorders. Created in BioRender. (2025) https://BioRender.com/skif5l4

Engineered antibodies, lipid nanoparticles, adeno-associated viruses and exosomes can be loaded with therapeutic protein, RNA or DNA cargo and directed to specific cell types within cardiac tissue (Momin et al., 2024). Cell-based therapies could also introduce engineered non-cardiomyocyte cells designed to restore proper electrical conduction.

New insights and technologies promise a new era of cardiac electrophysiology in which the heart's entire multicellular electrical ensemble, not just the cardiomyocyte, takes centre stage.

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

## Additional information

### Competing interests

No competing innterests declared.

### Author contributions

J.G and N.M jointly conceived and designed content, drafted text and prepared the figure. C.L and C.C edited and revised the manuscript. All authors have read and approved the final version of this manuscript sumimtted for publication, and agree to be accountable for all aspects of the work. All persons designated as authors qualify for authorship, and all those who qualify for authorship are listed.

### Funding

JG was supported by the German Centre for Cardiovascular Research (DZHK), German Research Foundation (DFG) – SFB1470-A4 and GR 5261/1-1, the German Society for Cardiology (DGK), Corona-Stiftung Grant (S199/10086/2022) and Dt Stiftung für Herzforschung/German Foundation of Heart Research (DSHF, F/37/22). CC was supported by was supported by ANR (ANR-23-CPJ1-0134-01) and the German Research Foundation (DFG) SFB1525 project number 453989101. NM was supported in part by awards from the American Heart Association (24IPA1269197), the WW Smith Charitable Trust, the PhRMA Foundation, the Penn Cardiovascular Institute Translational Dream Team Initiative and the Penn Center for Precision Engineering for Human Health.

### Acknowledgements

### Keywords

cardiac electrophysiology, non-myocytes, heterocellularity

### Supporting information

Additional supporting information can be found online in the Supporting Information section at the end of the HTML view of the article. Supporting information files available:

**Peer Review History**

