## [Peer Review History · The Journal of Physiology]

More Than Myocytes: The Heart's Electrical Ensemble

Jana Grune, Christof Lenz, Clément Cochain, and Noor Momin

DOI: 10.1113/JP289679

Corresponding author(s): Jana Grune (jana.grune@charite.de)

The following individual(s) involved in review of this submission have agreed to reveal their identity: Robert G. Gourdie (Referee #2)

Review Timeline:

Submission Date:	15-Jul-2025
Editorial Decision:	21-Jul-2025
Revision Received:	22-Jul-2025
Accepted:	24-Jul-2025

Senior Editor: Bjorn Knollmann

Reviewing Editor: T Alexander Quinn

Transaction Report:

Dear Dr Grune,

Re: JP-P-2025-289679 "**More Than Myocytes: The Heart's Electrical Ensemble**" by Jana Grune, Christof Lenz, Clément Cochain, and Noor Momin

Thank you for submitting your manuscript to The Journal of Physiology. It has been assessed by a Reviewing Editor and by 2 expert referees and we are pleased to tell you that it is acceptable for publication following satisfactory revision.

The review comments are copied at the end of this email.

Please address all the points raised and incorporate all requested revisions or explain in your Response to Referees why a change has not been made. We hope you will find the comments helpful and that you will be able to return your revised manuscript within 2 weeks. If you require longer than this, please contact journal staff: jp@physoc.org.

REVISION CHECKLIST:

We look forward to receiving your revised submission.

Yours sincerely,

Bjorn Knollmann
Senior Editor
The Journal of Physiology

EDITOR COMMENTS

Reviewing Editor:

This is a well written perspective, that requires only minor revisions based on the reviewers' comments, which suggest:

1) Providing more detail on the limitations and challenges described in the reviews, particularly regarding the specificity and translational hurdles of the discussed imaging and genetic techniques.

2) Inclusion of discussion about the role of other, less-discussed cell types (e.g., endothelial cells, neural inputs) in modulating conduction and their relevance in arrhythmogenesis.

3) Altering the excessively long list of institutional affiliations.

Senior Editor:

I concur with the recommendations made by the reviewing editor. Please make the list of institutional affiliations more concise and only name the primary institutions of the author.

REFEREE COMMENTS

Referee #1:

The paper offers a welcome set of comments, contextualising JP articles.

Referee #2:

The perspective is concise and distills the reviews' main concepts, particularly regarding the 2 salient paradigm shifts 1) from a myocyte-only view to a multicellular understanding, and 2) the growing understanding that intercellular electrical activity in the heart is not solely determined by electronic communication, but also ephaptic/electric field based mechanisms such as that embodied in the perinexus. The identification of future therapeutic strategies targeting non-myocytes is appropriately emphasized and presents the most innovative clinical implication arising from the reviews.

While the perspective references advanced imaging and genetic tools, it would benefit from more detail on the limitations and challenges, as described in the reviews, particularly regarding the specificity and translational hurdles of these technologies. The role of other, less-discussed cell types (e.g., endothelial cells, neural inputs) in modulating conduction is also underplayed (e.g. relative to immune cell types); the reviews dedicate space to these aspects and their relevance in arrhythmogenesis.

END OF COMMENTS

Response-to-Refereee**EDITOR COMMENTS****Reviewing Editor:**

This is a well written perspective, that requires only minor revisions based on the reviewers' comments, which suggest:

Thank you.

1) Providing more detail on the limitations and challenges described in the reviews, particularly regarding the specificity and translational hurdles of the discussed imaging and genetic techniques.

We agree with the reviewer, that the discussion of translational hurdles were cut short in our previous manuscript version. In light of the allowed word count, we have implemented one more text section to address the reviewers concern, which reads as follows:

“While these approaches have revolutionized the field, they also highlight the technical challenges that remain. Genetic targeting even with modern promoters to drive the expression of optogenetic tools can lack desired cell-type specificity needed to investigate different non-cardiomyocytes cell subsets. Computational models rely on assumed parameters that are sometimes difficult to empirically measure. Refinement in transgenic tools and implementation of artificial intelligence and machine learning into cardiac electrophysiology modeling promises to overcome some of these technical barriers and reveal even deeper insights into the heart's electrical complexity.”

2) Inclusion of discussion about the role of other, less-discussed cell types (e.g., endothelial cells, neural inputs) in modulating conduction and their relevance in arrhythmogenesis.

We agree with the reviewer's assessment and have implemented new text sections focusing on other non-myocyte cell types. The text reads as follows:

“Both computational models and experimental findings converge on the finding that these modes of electrical coupling are used by cardiac fibroblasts and immune cells like macrophages to actively support cardiomyocytes' electrical activity. These cells can speed up, slow down, or even block electrical conduction, fundamentally altering the heart's ability to function in both health and disease.

Beyond fibroblasts and immune cells, other non-cardiomyocytes in the heart also contribute to its electrical activity. Endothelial cells can also couple to cardiomyocytes via gap junctions, but whether such coupling occurs meaningfully in native tissue remains unclear. Additionally, the heart possesses many neurons, especially in arrhythmic niches, yet how neurons couple to cardiomyocytes is not well understood. Understanding the role of all cardiac cell types on cardiac electrophysiology in more detail is important.”

3) Altering the excessively long list of institutional affiliations.

We have shortened the number of affiliations to a maximum of two per author.

Senior Editor:

I concur with the recommendations made by the reviewing editor. Please make the list of institutional affiliations more concise and only name the primary institutions of the author.

We have shortened the number of affiliations to a maximum of two per author. It is unfortunately not possible to list only a single institution per author as some of our co-authors are running two independent labs or research groups at two independent institutions, so that there is no defined 'prototypic' primary institution. We hope that the editor agrees with this decision.

REFEREE COMMENTS

Referee #1:

The paper offers a welcome set of comments, contextualising JP articles.

Thank you.

Referee #2:

The perspective is concise and distills the reviews' main concepts, particularly regarding the 2 salient paradigm shifts 1) from a myocyte-only view to a multicellular understanding, and 2) the growing understanding that intercellular electrical activity in the heart is not solely determined by electronic communication, but also ephaptic/electric field based mechanisms such as that embodied in the perinexus. The identification of future therapeutic strategies targeting non-myocytes is appropriately emphasized and presents the most innovative clinical implication arising from the reviews.

Thank you.

While the perspective references advanced imaging and genetic tools, it would benefit from more detail on the limitations and challenges, as described in the reviews, particularly regarding the specificity and translational hurdles of these technologies. The role of other, less-discussed cell types (e.g., endothelial cells, neural inputs) in modulating conduction is also underplayed (e.g. relative to immune cell types); the reviews dedicate space to these aspects and their relevance in arrhythmogenesis.

We agree with the reviewer, that the discussion of translational hurdles were cut short in our previous manuscript version. In light of the allowed word count, we have implemented one more text section to address the reviewers concern, which reads as follows:

“While these approaches have revolutionized the field, they also highlight the technical challenges that remain. Genetic targeting even with modern promoters to drive the expression of optogenetic tools can lack desired cell-type specificity needed to investigate different non-cardiomyocytes cell subsets. Computational models rely on assumed parameters that are sometimes difficult to empirically measure. Refinement in transgenic tools and implementation of artificial intelligence and machine learning into cardiac electrophysiology modeling promises to overcome some of these technical barriers and reveal even deeper insights into the heart's electrical complexity.”

We agree with the reviewer's assessment and have implemented new text sections focusing on other non-myocyte cell types. The text reads as follows:

“Both computational models and experimental findings converge on the finding that these modes of electrical coupling are used by cardiac fibroblasts and immune cells like macrophages to actively support cardiomyocytes' electrical activity. These cells can speed up, slow down, or even block electrical conduction, fundamentally altering the heart's ability to function in both health and disease.

Beyond fibroblasts and immune cells, other non-cardiomyocytes in the heart also contribute to its electrical activity. Endothelial cells can also couple to cardiomyocytes via gap junctions, but whether such coupling occurs meaningfully in native tissue remains unclear. Additionally, the heart possesses many neurons, especially in arrhythmic niches, yet how neurons couple to cardiomyocytes is not well understood. Understanding the role of all cardiac cell types on cardiac electrophysiology in more detail is important.“

Dear Dr Grune,

Re: JP-P-2025-289679R1 "**More Than Myocytes: The Heart's Electrical Ensemble**" by Jana Grune, Christof Lenz, Clément Cochain, and Noor Momin

We are pleased to tell you that your paper has been accepted for publication in The Journal of Physiology.

Yours sincerely,

Bjorn Knollmann
Senior Editor
The Journal of Physiology

If you would like to receive our 'Research Roundup', a monthly newsletter highlighting the cutting-edge research published in The Physiological Society's family of journals (The Journal of Physiology, Experimental Physiology, Physiological Reports, The Journal of Nutritional Physiology, and The Journal of Precision Medicine: Health and Disease), please click this link, fill in your name and email address and select 'Research Roundup':

<https://www.physoc.org/journals-and-media/membernews>

- You can help your research get the attention it deserves! Check out Wiley's free Promotion Guide for best-practice recommendations for promoting your work at: www.wileyauthors.com/eeo/guide. You can learn more about Wiley Editing Services which offers professional video, design, and writing services to create shareable video abstracts, infographics, conference posters, lay summaries, and research news stories for your research at: www.wileyauthors.com/eeo/promotion.

The Corresponding Author will receive an email from Wiley with details on how to register or log-in to Wiley Authors Services where you will be able to place an order

EDITOR COMMENTS

Reviewing Editor:

Thank you for writing this perspective and for responding to the reviewer's comments.

Senior Editor:

Thank you for your excellent contribution!

REFeree COMMENTS

Referee #2:

Congratulations to the authors on their thoughtful editorial.